# Association of Non-Invasive Positive Pressure Ventilation with Short-Term Clinical Outcomes in Patients Hospitalized for Acute Decompensated Heart Failure

**DOI:** 10.3390/jcm10215092

**Published:** 2021-10-29

**Authors:** Midori Yukino, Yuji Nagatomo, Ayumi Goda, Takashi Kohno, Makoto Takei, Yosuke Nishihata, Mike Saji, Yuichi Toyosaki, Shintaro Nakano, Yukinori Ikegami, Yasuyuki Shiraishi, Shun Kohsaka, Takeshi Adachi, Tsutomu Yoshikawa

**Affiliations:** 1Department of Cardiology, National Defense Medical College, Tokorozawa 359-8513, Japan; doc33079@ndmc.ac.jp (M.Y.); con467@ndmc.ac.jp (Y.I.); tadachibu@gmail.com (T.A.); 2Department of Cardiovascular Medicine, Kyorin University School of Medicine, Tokyo 181-8611, Japan; ayumix34@yahoo.co.jp (A.G.); kohno-ta@ks.kyorin-u.ac.jp (T.K.); 3Department of Cardiology, Saiseikai Central Hospital, Tokyo 108-0073, Japan; makoto_tk@hotmail.com; 4Department of Cardiology, St. Luke’s International Hospital, Tokyo 104-8560, Japan; hatasuke@luke.ac.jp; 5Department of Cardiology, Sakakibara Heart Institute, Tokyo 183-0003, Japan; mikesaji8@gmail.com (M.S.); tyoshi@shi.heart.or.jp (T.Y.); 6Department of Cardiology, Saitama Medical University International Medical Center, Hidaka 350-1298, Japan; toyoyuki6465@yahoo.co.jp (Y.T.); snakano@saitama-med.ac.jp (S.N.); 7Department of Cardiology, National Hospital Organization, Tokyo Medical Center, Tokyo 152-8902, Japan; 8Department of Cardiology, Kyorin University Faculty of Medicine, Tokyo 160-8582, Japan; white_cascade_libra@yahoo.co.jp (Y.S.); sk@keio.jp (S.K.)

**Keywords:** NPPV, acute decompensated heart failure, ischemic heart disease, intensive care, endotracheal intubation, length of hospital stay

## Abstract

The real-world evidence has been sparse on the impact of non-invasive positive pressure ventilation (NPPV) on the outcomes in acute decompensated heart failure (ADHF) patients. We aim to explore this issue in the prospective multicenter WET-HF registry. Among 3927 patients (77 (67–84) years, male 60%), the NPPV was used in 775 patients (19.7%). The association of NPPV use with in-hospital outcome and length of hospital stay (LOS) was examined by two methods, propensity score (PS) matching and multivariable analysis with adjustment for PS. In these analyses the NPPV group exhibited a lower endotracheal intubation (ETI) rate and a comparable in-hospital mortality, but longer LOS compared to the non-NPPV group. In the stratified analysis, the NPPV group exhibited a significantly lower ETI rate in patients with ischemic etiology, systolic blood pressure (sBP) > 140 mmHg and the Controlling Nutritional Status (CONUT) score ≤ 3, indicating better nutritional status. On the contrary, NPPV use was associated with longer LOS in patients with non-ischemic etiology, sBP < 100 mmHg and CONUT score > 3. In conclusion, NPPV use was associated with a lower incidence of ETI. Particularly, patients with ischemic etiology, high sBP, and better nutritional status might benefit from NPPV use.

## 1. Introduction

Heart failure (HF) is a major social problem that has been increasing in prevalence worldwide due to a rapidly aging society [1]. The current pathophysiologic understanding of acute decompensated HF (ADHF) is incomplete. Due to the lack of adequately conducted trials to address the unmet need for evidence therapy in ADHF, the guideline recommendations for the management of ADHF are based only on algorithms provided by expert consensus guided by blood pressure and/or clinical signs of congestion or hypoperfusion [2]. Non-invasive positive pressure ventilation (NPPV) has been used to treat acute exacerbations of chronic respiratory diseases instead of traditional endotracheal intubation (ETI) since 1990 [3]. Over the past two decades, NPPV has been increasingly used in patients with acute cardiogenic pulmonary edema (ACPE) [4]. NPPV improves oxygenation, decreases breathing effort [4], and reduces left ventricular afterload [4] and both right and left ventricular preload [5] in patients with ACPE. Prompt improvement in patient-reported dyspnea, acidosis, hypercapnia, and tachycardia has been consistently reported after NPPV use [6].

The small randomized control trials (RCTs) exploring the impact of NPPV using continuous positive airway pressure (CPAP) or bilevel positive airway pressure (BiPAP) on the outcomes have been conducted by different research groups [6,7,8,9]. In previously conducted RCTs, several significant issues have been indicated, such as a small number of participants in most studies, lack of blinding and possible publication biases. Even the 3CPO trial [6], one of the largest RCTs, failed to demonstrate the beneficial effect of NPPV on short-term mortality. In that study, significant overlap of treatment arms (standard oxygen therapy, CPAP, and NPPV) biased the findings of the outcome, which suggests the difficulty of appropriate randomization in an urgent care setting. Furthermore, the recruitment of patients included in these RCTs was highly selective and the study populations in most of those studies were relatively younger compared to the patients with acute decompensated heart failure (ADHF) in recent years [10,11]. Thus, it remains uncertain whether the findings in these trials can be applicable for the contemporary ADHF population in a real-world setting.

Regarding the impact of NPPV on clinical outcomes, recent meta-analysis [10] demonstrated that NPPV might reduce the need for ETI and in-hospital mortality, although the findings of RCTs were not consistent regarding its impact on mortality [6,7,8,9]. However, meta-analysis potentially includes uncontrolled biases due to the lack of description of standard procedures (diuretics, vasodilators, and catecholamines) and the differences in study protocols (intubation criteria, etc.). Further, previous reports including meta-analysis did not fully identify which patient groups would benefit most from NPPV, although one small RCT has shown the beneficial effect of NPPV in patients with acute myocardial infarction-associated pulmonary edema that is not complicated by shock [12]. Hence, we aimed to examine the impact of NPPV on short-term clinical outcomes and to further explore the patient groups who would benefit from NPPV use in the multicenter West Tokyo Heart Failure (WET-HF) registry.

## 2. Materials and Methods

### 2.1. Study Design

We analyzed data from 4000 ADHF patients registered in the WET-HF registry from 2006 to 2017. The WET-HF registry is a multicenter prospective cohort registry enrolling all patients hospitalized for ADHF according to the Framingham criteria [13]. In this registry, patients with acute coronary syndrome or isolated right-sided HF were excluded. The clinical diagnosis of ADHF was made by individual cardiologists at each institution. The eight study centers were located in Tokyo, Japan, and included four university hospitals (Keio University, Kyorin University, Saitama Medical University, and National Defense Medical College) and four tertiary referral hospitals (Sakakibara Heart Institute, St. Luke’s International Hospital, Saiseikai Central Hospital, and National Hospital Organization Tokyo Medical Center) [14,15,16].

Baseline data and outcomes for the WET-HF registry were collected by dedicated clinical research coordinators from medical records and interviews with treating physicians to obtain a robust assessment of the care and patient outcomes. Data regarding NPPV use and ETI were also prospectively collected. Data were entered into an electronic data-capturing system with a robust data query engine and system validations for data quality. Outliers in continuous variables or unexpected values in the categorical variables were selected based on established criteria, and the originating institution was notified to verify the values. The quality of reporting was also verified by the principal investigators (Y.S. and S.K.) at least once a year, and periodic queries were conducted to ensure quality. Exclusive on-site auditing by the investigators (Y.S. and S.K.) ensured proper registration of each patient. Before the launch of the WET-HF registry, information regarding the objective of the present study, its social significance, and an abstract were provided for clinical trial registration with the University Hospital Medical Information Network (UMIN000001171).

The study protocol was approved by the institutional review boards at each hospital, and the study was conducted in accordance with the principles of the Declaration of Helsinki. Written and/or oral informed consent was obtained from each subject before registration.

Figure 1 shows the flowchart of the present study. Within the cohort, we excluded 73 patients who lacked information regarding NPPV use during the index admission. The remaining 3927 patients (77 (67–84) years, 60% men) were included in the analysis (pre-matched cohort). We categorized them into two groups: patients who received NPPV and those who did not receive NPPV. Additionally, the propensity score was calculated using variables such as age, sex, year of admission, etiology, comorbidities, New York Heart Association (NYHA) functional class, vital signs, symptoms, laboratory data, and pre-admission medication. One-to-one nearest-neighbor propensity matching was conducted between patients from the NPPV group and those from the non-NPPV group, which resulted in 433 pairs available for analysis as the post-matched cohort.

### 2.2. Endpoint

In the WET-HF registry, information regarding ETI during the index admission was prospectively collected. A follow-up survey using medical charts or telephone reviews was performed, and patients who were lost to follow-up were censored at the date of last contact. Information regarding specific outcomes was obtained from the participating cardiologists and investigators. This information included all-cause mortality, readmission for ADHF, and a composite of all-cause mortality and readmission for ADHF.

### 2.3. Statistical Analysis

Continuous variables were expressed as mean ± standard deviation for normally distributed data and as median (interquartile range) for data with non-normal distribution. Between-group differences were assessed with an unpaired t-test or the Mann–Whitney U test for the unpaired data, while the chi-squared test was used for the comparison of discrete variables. Kaplan–Meier survival curves were constructed for each group, and differences between the groups were analyzed using the log-rank test.

The propensity score was developed using the clinical variables listed in Table 1. The variables were selected a priori for their potential to be strongly associated with NPPV use. For the multivariable analysis, age, sex, left ventricular ejection fraction (LVEF) and the variables that showed an association with NPPV use in the univariate analysis with *p* value < 0.1 were employed. The propensity score was then derived using a generalized logistic model to predict the probability of receiving NPPV. The logit of this score was utilized with a caliper of 0.2 to obtain the propensity score for the matching process. Patients from the NPPV and non-NPPV groups were matched using propensity score methods. Greedy nearest-neighbor matching was performed sequentially in a 1:1 fashion without replacement. We calculated the standardized differences to assess bias between the groups. The balance on potential baseline confounders was evaluated using standardized differences, with an importance threshold set at 0.10 a priori.

Additional analysis was conducted in order to further confirm the association of NPPV use with ETI, in-hospital mortality and length of hospital stay (LOS). Multivariable logistic regression analysis was conducted for each endpoint with adjustment for calculated propensity score in the pre-matched cohort. Multiple regression analysis was conducted to see the association of NPPV use with LOS with adjustment for calculated propensity score in the pre-matched cohort.

We performed a stratified analysis to determine the patient population that would benefit from the use of NPPV. Older age, hypotension, malnutrition [17], and history of ADHF hospitalization [18] are risk factors for adverse outcome in patients with HF. Thus, in addition to basic information such as gender, BMI, and LVEF, the patients were stratified by the factors potentially affecting the effect of NPPV use such as age, sBP, nutritional status and history of ADHF hospitalization. We used the Controlling Nutritional Status (CONUT) score to assess nutritional status. This is a screening tool for the nutritional status of hospitalized patients [19], which is calculated from serum albumin, total cholesterol, and lymphocyte count. These were measured during the index hospitalization in the present study. Interactions between the subgroups were tested and when p value was <0.1, for each stratified group logistic regression analysis was conducted for each endpoint with adjustment for propensity score. Similarly, when p value for interaction was <0.1, multiple regression analysis was conducted to see the association of NPPV use with LOS with adjustment for propensity score.

Statistical significance was set at *p* < 0.05. All statistical analyses were performed using JMP 14.2.0 (SAS Institute, Cary, NC, USA).

## 3. Results

### 3.1. Baseline Characteristics

In the pre-matched cohort, 775 (19.7%) patients received NPPV. Baseline characteristics of the NPPV and non-NPPV groups in the pre-matched cohort are presented in Appendix A.

### 3.2. Factors Associated with NPPV Use

Multivariable logistic regression analysis was conducted to identify the predictors of NPPV use (Table 1). Admission in the third admission period (2014–2017), ischemic etiology, higher systolic blood pressure (sBP), and parameters representing severe ADHF such as lower oxygen saturation (SpO_2_), NYHA class IV, and higher B-type natriuretic peptide (BNP)/N-terminal pro B-type natriuretic peptide (NT-pro BNP) were significantly associated with NPPV use. Using these results, the propensity score was calculated and one-to-one nearest-neighbor propensity matching was conducted (*n* = 433 in each group) to adjust for confounding factors such as severity of HF and in-hospital treatment between the groups (Figure 1).

### 3.3. Findings in the Post-Matched Cohort

In the post-matched cohort, no significant differences were observed in the majority of the baseline characteristics between the NPPV and non-NPPV groups, and the standardized differences were mostly within 0.1 (Table 2). The NPPV group exhibited a lower ETI rate compared to the non-NPPV group (Figure 2A) and comparable in-hospital mortality (Figure 2B) during admission, but LOS was longer in the NPPV group (Figure 2C), as observed in the pre-matched cohort. Since there was a difference in in-hospital treatment between NPPV and non-NPPV groups, multiple regression analysis was conducted. NPPV use was not associated with longer LOS after adjustment of the other in-hospital treatment (β = 0.71, standard error of the mean [SEM] 0.73, *t*-value 0.98, *p* = 0.33)

### 3.4. Sensitivity Analysis

Additional analysis was conducted in order to further confirm the association of NPPV use with ETI, in-hospital mortality and LOS. Logistic regression analysis was conducted for each endpoint with adjustment for calculated propensity score in the pre-matched cohort. NPPV use was associated with lower incidence of ETI after adjustment for propensity score (Figure 3A). However, NPPV use was not associated with in-hospital death (Figure 3B). Multiple regression analysis revealed that NPPV use was associated with longer LOS (Table 3).

### 3.5. Stratified Analysis

The stratified analysis was conducted. For ETI rate, interactions between the subgroups were seen in etiology (ischemic or non-ischemic), sBP, LVEF, and CONUT score (Figure 3C). In ischemic etiology, sBP > 140, LVEF < 50%, and CONUT score < 3 (indicating better nutritional status), the NPPV group exhibited a lower ETI rate (Figure 3C). For in-hospital mortality, only the history of ADHF admission modified the effect of NPPV. But NPPV use was not associated with significant changes in in-hospital mortality, either in patients with previous ADHF admission or those without (Figure 3D). For LOS, interactions between the subgroups were seen in etiology (ischemic or non-ischemic), previous ADHF admission, sBP and CONUT score. In contrast to the findings of ETI rate, the NPPV group was associated with a longer LOS in the following subgroups: non-ischemic etiology, no previous ADHF admission, sBP < 100 and CONUT score > 3 (Table 3).

## 4. Discussion

In the present study, we demonstrated the following main findings: (1) In the post-matched cohort, NPPV use was associated with a lower ETI rate, but there were no differences in in-hospital mortality during admission. NPPV use was associated with slightly longer LOS, but it was not statistically significant after adjustment for in-hospital treatment. (2) Lower ETI rate, no effect in in-hospital mortality and longer LOS in NPPV group were further confirmed by the analysis in the pre-matched cohort with adjustment for propensity score. (3) NPPV use was associated with a lower ETI rate in some subgroups such as patients with ischemic etiology, sBP > 140 mmHg at admission, LVEF < 50%, and better nutritional status indicated by CONUT score ≤ 3. (4) NPPV use was associated with a longer LOS in patients with non-ischemic etiology, no history of ADHF admission, sBP < 100 mmHg, and poorer nutritional status indicated by CONUT score >3.

These findings suggest that NPPV use might be associated with benefits such as avoidance of ETI. Furthermore, some subgroups such as patients with ischemic etiology might benefit from NPPV but, on the contrary, others might experience disadvantages such as longer LOS along with receiving NPPV. The strength of our study is that we were able to eliminate bias by analysis using propensity score to align patient backgrounds. Moreover, stratified analysis clearly suggest the subgroups that showed advantages and/or disadvantages along with receiving NPPV in a real-world setting.

### 4.1. Impact of NPPV Use on ACPE

The clinical use of NPPV has been increasing since the 1980s, and small RCTs have evaluated the efficacy of NPPV in ACPE [6,7,8,9,10,20]. Several studies have shown the benefit of NPPV in terms of reduced ETI rate, but findings regarding the impact of NPPV on in-hospital mortality had been inconsistent [8,9,17,18]. In this context, the recommendations regarding the use of NPPV in ACPE are inconsistent between the European Society of Cardiology guidelines (class IIa) [21] and the American College of Cardiology/American Heart Association guidelines, which do not provide treatment guidance for ACPE [22]. According to a recent meta-analysis of RCTs analyzing the use of NPPV to treat ACPE, NPPV use was associated with reduced in-hospital mortality as well as ETI rate [10].

The results of the present study are in line with the results published to date in terms of reduced ETI. Furthermore, a risk ratio for ETI in the propensity match analysis was 0.515 and an estimated risk ratio calculated from OR in the multivariable regression analysis [23] was 0.480 (Figure 2A and Figure 3A). From these findings the effect sizes in our study were also consistent with the recent meta-analysis (risk ratio 0.49) [10].

The lack of benefit in terms of in-hospital mortality might be due to the lower overall mortality in the present cohort. Although most of the previous studies reported in-hospital mortality exceeding 10% [10,24,25,26], it was as low as 4% in the present study (Figure 2).

### 4.2. The Relationship among Ischemic Etiology, Hypertension, and NPPV Use

In the present study, ischemic etiology and higher sBP were associated with NPPV use (Table 2). Hypertension is closely associated with impaired relaxation [27] and stiffness [28] of the left ventricular myocardium. The combination of physiological and/or psychological stress causes vasoconstriction through neurohormonal activation, leading to an increase in the left atrial pressure. This continuum of responses causes further distress, resulting in rapid development of “flash” pulmonary edema [29]. Ischemic etiology [15] and higher aortic stiffness [30,31] are also among the factors related to this pathology.

On the other hand, the stratified analysis revealed that NPPV use might provide benefits, especially for patients with ischemic etiology and sBP > 140 mmHg, in terms of reduced ETI rate. Quick application of NPPV in patients with ADHF might be advantageous for those with rapidly progressing oxygen desaturation caused by the aforementioned mechanism. We propose reduced right and left ventricular preload [32] and LV afterload due to decreased transmural (or transthoracic) pulmonary pressure [33] by positive pressure ventilation as another potential mechanism mediating the benefit of NPPV in these subgroups. A positive association was observed between elevated left ventricular end-diastolic pressure and subendocardial ischemia [34]. In addition, NPPV promptly improves oxygenation, which can ameliorate myocardial ischemia [4].

### 4.3. The Subgroups Associated with Longer LOS along with Receiving NPPV

In the present study, LOS was longer in the NPPV group even in the post-matched cohort (Figure 2C), although it was not statistically significant after adjustment for other in-hospital treatment. Recent meta-analysis did not show a statistically significant difference in LOS between NPPV and non-NPPV groups [10]. However, the recent finding from the registry showed prolonged LOS in NPPV group even after propensity matching [26]. Collectively, those findings including the present study might suggest the association of NPPV use with longer LOS exclusively in real-world settings. NPPV use was associated with prolonged LOS, especially in females and in patients with age ≥ 75 years, non-ischemic etiology, and poorer nutritional status indicated by CONUT score > 3. Interestingly, in some of the subgroups, one group showed the benefit of NPPV in terms of ETI avoidance, while the other was associated with longer LOS (e.g., age ≥75 vs. <75, non-ischemic vs. ischemic, CONUT score ≤3 vs. >3). Previous studies have reported that older age, female sex [35], and poorer nutritional status [36] were associated with frailty, which in turn was associated with a high risk for short-term adverse outcomes and longer LOS [37]. These patient groups might be prone to complications or deconditioning during or after NPPV application.

### 4.4. Limitations

The present study has some limitations. This study was based on observational registry data. Despite covariate adjustment using propensity matching, unmeasured or unknown variables might have influenced the outcomes. Since the deviation of in-hospital treatment remained even after post-matched cohort, it might have influenced our findings.

The small number of in-hospital events did not allow us to conduct the multivariable analysis due to the concern for overfitting, especially in the stratified analysis. The decision to use NPPV was made by the attending physician, resulting in selection bias. Information regarding the timing, situation (emergency department, intensive care unit, or general ward), duration of NPPV use, mode of NPPV (CPAP or BIPAP), setting (the pressure at inspiration and expiration), and the brand of NPPV equipment was not collected. Data regarding the details of adverse events related to NPPV use were not collected. Since this study involved only Japanese subjects, our findings might not be applicable to other countries due to factors such as a unique insurance system and long hospital stay. Increased frequency of NPPV use over time might suggest a learning curve effect and the findings need to be interpreted with caution, although the year of admission was adjusted in the propensity matching process.

## 5. Conclusions

The NPPV group showed comparable in-hospital outcomes with the non-NPPV group within the matched cohort. However, NPPV might prevent ETI in ADHF patients with respiratory distress. Particularly, patients with younger age, coronary artery disease, and better nutritional status might benefit from NPPV use. On the contrary, we need caution when applying NPPV for the patients with features related to frailty such as elderly, female, or poorer nutritional status, since it might lead to longer LOS.

## Figures and Tables

**Figure 1 jcm-10-05092-f001:**
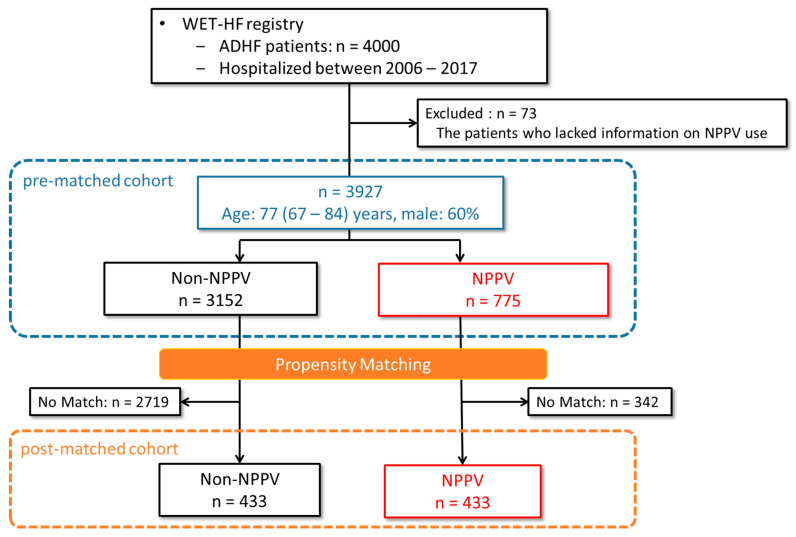
Flowchart of the study. Altogether, 4000 ADHF patients enrolled in WET-HF registry were divided into 2 groups, namely patients who received NPPV and those who did not receive NPPV in the pre-matched cohort. After propensity matching, 433 pairs were available as the post-matched cohort. ADHF: acute decompensated heart failure, WET-HF: West Tokyo Heart Failure, NPPV: non-invasive positive pressure ventilation.

**Figure 2 jcm-10-05092-f002:**
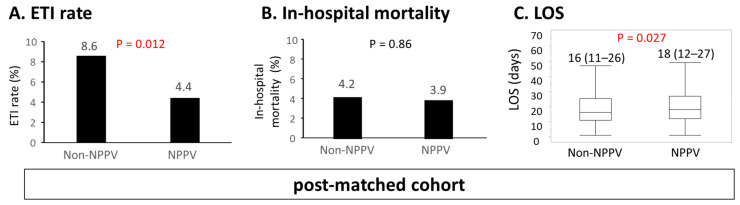
In-hospital outcomes in the post-matched cohort. (**A**) ETI rate, (**B**) in-hospital mortality, and (**C**) LOS in the post-matched cohort are depicted. ETI, endotracheal intubation; NPPV, non-invasive positive pressure ventilation; LOS, length of hospital stay.

**Figure 3 jcm-10-05092-f003:**
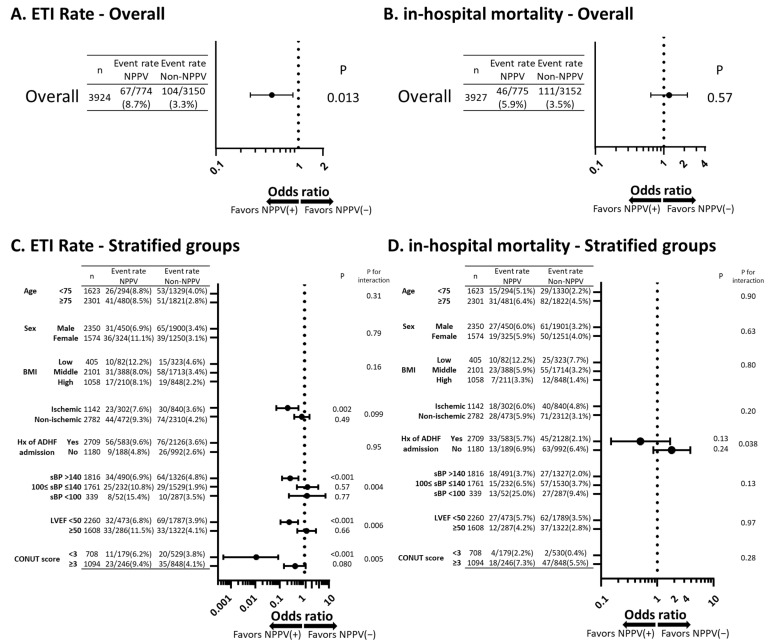
In-hospital outcomes with adjustment for propensity score in the pre-matched overall cohort and the stratified groups. (**A**) ETI rate for overall population, (**B**) in-hospital mortality for overall population, (**C**) ETI rate for stratified groups, and (**D**) in-hospital death for stratified groups are depicted. ETI, endotracheal intubation; NPPV, non-invasive positive pressure ventilation; BMI, body mass index (low BMI: BMI < 18.5, middle BMI: 18.5 ≤ BMI < 25, high BMI: BMI ≥ 25); Hx, history; sBP, systolic blood pressure; LVEF, left ventricular ejection fraction; CONUT, the Controlling Nutritional Status.

**Table 1 jcm-10-05092-t001:** Univariable and multivariable logistic regression for non-invasive positive pressure ventilation Use.

	Univariable			Multivariable		
	Odds Ratio	95% CI	*p* Value	Odds Ratio	95% CI	*p* Value
Age	1.006	1.000~1.012	0.050	0.998	0.988~1.008	0.74
Sex (female)	1.097	0.935~1.287	0.25	1.045	0.819~1.334	0.72
Admission year 2006–2009	1 (ref)	-	-	1 (ref)	-	-
2010–2013	5.726	3.630~3.033	<0.001	3.660	1.962~6.829	<0.001
2014–2017	7.368	4.704~11.542	<0.001	6.389	3.466~11.76	<0.001
Etiology ICM	1 (ref)	-	-	1 (ref)	-	-
DCM	0.369	0.275~0.494	<0.001	0.461	0.300~0.707	<0.001
VHD	0.598	0.486~0.735	<0.001	0.866	0.634~1.183	0.37
Prior ADHF admission	0.695	0.581~0.833	<0.001	0.898	0.584~0.950	0.45
Atrial fibrillation	0.592	0.504~0696	<0.001	0.745	0.584~0.950	0.018
Home oxygen therapy	1.684	1.139~2.488	0.006	1.573	0.869~2.847	0.14
Dialysis	2.484	1.735~3.555	<0.001	1.704	0.842~3.449	0.14
sBP	1.016	1.014~1.018	<0.001	1.012	1.009~1.016	<0.001
Heart Rate	1.014	1.011~1.016	<0.001	1.009	1.005~1.013	<0.001
SpO_2_	0.911	0.899~0.923	<0.001	0.914	0.898~0.931	<0.001
NYHA (IV/II–III)	2.806	2.380~3.308	<0.001	2.292	1.806~2.909	<0.001
Cold extremities	2.777	2.290~3.368	<0.001	2.196	1.714~2.814	<0.001
Rales	2.589	2.172~3.087	<0.001	1.554	1.221~1.977	<0.001
BNP/NT-proBNP quartile 1st	1 (ref)	-	-	1 (ref)	-	-
2nd	1.329	1.040~1.700	0.023	1.215	0.864~1.708	0.26
3rd	1.495	1.174~1.904	0.001	1.175	0.789~1.582	0.53
4th	2.128	1.686~2.686	<0.001	1.191	0.822~1.725	0.35
CRP	13.421	6.292~28.703	0.075	1.075	1.042~1.109	<0.001
eGFR	0.334	0.154~0.713	0.005	0.999	0.993~1.004	0.63
LVEF	0.821	0.534~1.263	0.37	0.997	0.993~1.007	0.58
Prehospital: β-blocker	0.751	0.639~0.882	<0.001	1.12	0.87~1.43	0.80
Prehospital: RASi	1.220	1.029~1.447	0.021	1.387	1.100~1.749	0.006
Prehospital: MRA	0.556	0.433~0.714	<0.001	0.764	0.530~1.099	0.14
Prehospital: loop diuretics	0.632	0.537~0.743	<0.001	0.914	0.709~1.179	0.49

CI, confidence interval; ref, reference; ICM, ischemic cardiomyopathy; DCM, dilated cardiomyopathy; VHD, valvular heart disease; ADHF, acute decompensated heart failure; sBP, systolic blood pressure; SpO_2_ oxygen saturation; NYHA, New York Heart Association; BNP, B-type natriuretic peptide; NT-pro BNP, N-terminal pro B-type natriuretic peptide; CRP, C-reactive protein; eGFR, estimated glomerular filtration rate; LVEF, left ventricular ejection fraction; RASi, renin-angiotensin system inhibitor; MRA, mineralocorticoid receptor antagonist.

**Table 2 jcm-10-05092-t002:** Baseline characteristics according to non-invasive positive pressure ventilation use in the post-matched cohort.

	Post-Matched Cohort			
	Non-NPPV (*n* = 433)	NPPV (*n* = 433)	*p* Value	SDM
Year			0.32	
2006–2009 (%)	20 (5)	15 (3)		0.01
2010–2013 (%)	142 (33)	127 (29)		0.01
2014–2017 (%)	271 (63)	291 (67)		0.01
Age (years)	77 (67–84)	78 (69–84)	0.40	0.06
Female (%)	183 (42)	172 (40)	0.45	0.04
BMI	23.0 (20.5–25.9)	23.1 (20.3–26.0)	0.74	0.03
Etiology			0.97	
DCM (%)	44 (10)	41 (9)		0.02
ICM (%)	157 (36)	156 (36)		0.01
VHD (%)	122 (28)	121 (28)		0.01
HFr/mr/pEF			0.82	
HFrEF (%)	169 (39)	175 (40)		0.02
HFmrEF (%)	85 (20)	88 (20)		0.02
HFpEF (%)	179 (41)	170 (39)		0.04
Prior ADHF admission (%)	116 (27)	118 (27)	0.88	0.01
Atrial Fibrillation (%)	179 (41)	186 (43)	0.63	0.03
Hypertension (%)	303 (70)	313 (72)	0.45	0.05
Dyslipidemia (%)	157 (36)	196 (45)	0.008	0.18
DM (%)	152 (35)	161 (37)	0.52	0.04
Smoking (%)	190 (44)	190 (44)	1	0
Dialysis (%)	17 (4)	16 (4)	0.86	0.01
COPD (%)	31 (7)	17 (4)	0.037	0.14
Stroke/TIA (%)	63 (15)	69 (16)	0.58	0.01
Home oxygen therapy (%)	22 (5)	20 (5)	0.75	0.02
Pacemaker (%)	28 (6)	27 (6)	0.89	0.01
ICD (%)	11 (3)	13 (3)	0.68	0.03
CRT (%)	3 (1)	4 (1)	0.70	0.03
Clinical Presentation at Admission				
NYHA II/III/IV			0.37	
II (%)	47 (11)	60 (14)		0.09
III (%)	105 (24)	107 (25)		0.01
IV (%)	271 (65)	266 (61)		0.07
sBP (mmHg)	147 (124–172)	147 (124–174)	0.93	0
Heart rate (bpm)	98 (78–118)	100 (81–120)	0.12	0.07
SpO_2_ (%)	95 (90–97)	94 (89–97)	0.39	0.002
PND (%)	193 (45)	222 (52)	0.029	0.15
Orthopnea (%)	252 (59)	272 (66)	0.056	0.13
Rales (%)	305 (70)	297 (69)	0.55	0.04
Sound III (%)	194 (45)	212 (50)	0.23	0.08
JVD (%)	217 (52)	213 (52)	0.91	0.01
Edema (%)	287 (66)	271 (63)	0.32	0.07
Cold extremities (%)	150 (35)	145 (33)	0.72	0.02
Laboratory Data				
BNP/NT-pBNP quartile			0.92	
1st	78 (18)	85 (20)		0.04
2nd	104 (24)	104 (24)		0.001
3rd	123 (28)	123 (28)		0.01
4th	128 (30)	121 (28)		0.04
BNP (pg/mL)	889 (447–1419)	871 (432–1575)	0.97	0.08
NT-proBNP (pg/mL)	4884 (2577–10,017)	4485 (2356–9239)	0.70	0.09
Hemoglobin (g/dL)	12.0 (10.4–13.8)	11.9 (10.5–13.7)	0.54	0.07
BUN (mg/dL)	22.6 (17.8–32.7)	22.0 (16.9–33.8)	0.93	0.04
eGFR (mL/min/1.73 m^2^)	46.5 (32.9–61.0)	48.6 (30.4–63.0)	0.57	0.05
UA (mg/dL)	6.7 (5.6–8.2)	6.4 (5.2–7.9)	0.043	0.14
Na (mEq/L)	140 (137–142)	140 (137–142)	0.55	0.1
CRP (mg/dL)	0.7 (0.2–2.4)	0.6 (0.2–2.5)	0.58	0.03
Echocardiography				
LVDd (mm)	52 (45–59)	52 (46–58)	0.97	0.003
LVEF (%)	45 (32–57)	44 (31–58)	0.87	0.01
LAD (mm)	43 (38–49)	43 (38–48)	0.36	0.04
E/e’	18.3 (13.2–27.5)	19.2 (13.9–26.6)	0.47	0.08
TRPG (mmHg)	29 (22–37)	29 (23–38)	0.45	0.03
Pre-hospital treatment				
ACE-I/ARB (%)	203 (47)	208 (48)	0.73	0.02
β-blocker (%)	187 (43)	182 (42)	0.73	0.02
MRA (%)	62 (14)	48 (11)	0.15	0.097
Loop diuretic(po.) (%)	170 (39)	171 (39)	0.94	0.004
Thiazide (%)	16 (4)	20 (5)	0.49	0.05
In-hospital treatment				
Loop diuretics (iv.) (%)	313 (72)	343 (79)	0.017	0.03
Nitrates (%)	129 (39)	197 (46)	<0.001	0.3
Carperitide (%)	210 (49)	241 (56)	0.032	0.15
PDE-III (%)	14 (3)	11 (3)	0.55	0.04
Catecholamine(%)	61 (14)	102 (24)	<0.001	0.24
IABP (%)	18 (4)	12 (3)	0.27	0.08

Categorical values are expressed as percentage, and continuous variables are expressed as median (interquartile range). NPPV, non-invasive positive pressure ventilation; SDM, standardized difference in means; BMI, body mass index; DCM, dilated cardiomyopathy; ICM, ischemic cardiomyopathy; VHD, valvular heart disease; HFrEF, HF with reduced ejection fraction; HFmrEF, HF with mid-range ejection fraction; HFpEF, HF with preserved ejection fraction; NYHA, New York Heart Association functional class; ADHF, acute decompensated heart failure; DM, diabetes mellitus; COPD, chronic obstructive pulmonary disease; TIA, transient ischemic attacks; ICD, implantable cardioverter defibrillator; CRT, cardiac resynchronization therapy; sBP, systolic blood pressure; SpO2, oxygen saturation; PND, paroxysmal nocturnal dyspnea; JVD, Jugular venous distention; BNP, B-type natriuretic peptide; NT-pro BNP, N-terminal pro B-type natriuretic peptide; BUN, blood urea nitrogen; Cr, serum creatinine; eGFR, estimated glomerular filtration rate; UA uric acid; Na, serum sodium; T Bil, total bilirubin; CRP, C-reactive protein; LVDd, left ventricular end-diastolic diameter; LVDs, left ventricular end-systolic diameter; LAD, left atrial diameter; TRPG, tricuspid regurgitation pressure gradient; ACEI, angiotensin-converting enzyme inhibitor; ARB, angiotensin receptor blocker; MRA, mineralocorticoid antagonist; PDE-III, phosphodiesterase-III inhibitor; IABP, intra-aortic balloon pumping.

**Table 3 jcm-10-05092-t003:** Multiple regression analysis for LOS with adjustment for propensity score in the pre-matched overall cohort and stratified groups.

		*n* (NPPV/Non-NPPV)	LOS (NPPV/Non-NPPV)	*p* for Interaction	*β*	SEM	*t*-Value	*p* Value
Overall		775/3152	14 (10–27)/14 (10–23)	NA	1.527	0.715	2.14	0.032
Stratified groups								
Age	<75	294/1330	15 (10–26)/15 (10–23)	0.21				
	≥75	481/1822	18 (11–27)/14 (9–22)					
sex	Male	450/1901	16 (10–26)/14 (10–23)	0.69				
	Female	325/1251	19 (11–29)/14 (9–22)					
BMI	Low	82/323	16 (9–28)/15 (9–28)	0.73				
	Middle	388/1714	17 (11–27)/14 (10–22)					
	High	211/848	17 (10–26)/14 (9–21)					
Etiology	Ischemic	302/840	17 (10–26)/15 (10–23)	0.008	0.069	1.217	0.06	0.96
	Non-ischemic	473/2312	17 (11–27)/14 (9–22)		2.375	0.88	2.69	0.007
Hx of ADHF admission	Yes	189/992	16 (9–26)/14 (10–24)	0.084	1.404	1.646	0.85	0.39
	No	583/2128	17 (11–27)/14 (9–22)		1.524	0.754	2.02	0.044
sBP	≥140	491/1327	16 (10–24)/14 (9–21)	0.062	1.362	0.901	1.51	0.13
	100≤ <140	232/1530	19 (12–31)/14 (10–23)		0.548	1.25	0.44	0.66
	<100	52/287	25 (12–40)/17 (10–31)		7.077	2.338	3.03	0.003
LVEF	<50	473/1789	17 (10–27)/15 (10–24)	0.50				
	≥50	287/1322	18 (11–27)/13 (9–21)					
CONUT score	≤3	179/530	14 (9–26)/13 (9–21)	0.044	0.523	0.919	0.57	0.57
	>3	248/848	20 (12–35) /16 (10–27)		4.044	1.306	3.1	0.002

LOS, length of hospital stay; NPPV, non-invasive positive pressure ventilation; SEM, standard error of the mean; BMI, body mass index; ADHF, acute decompensated heart failure; Hx, history; sBP, systolic blood pressure; LVEF, left ventricular ejection fraction; CONUT, the Controlling Nutritional Status.

## Data Availability

The data underlying this article cannot be shared publicly to maintain the privacy of individuals that participated in the study. The data will be shared on reasonable request to the corresponding author.

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
