# Peer review of "Association of Non-Invasive Positive Pressure Ventilation with Short-Term Clinical Outcomes in Patients Hospitalized for Acute Decompensated Heart Failure"

_jcm, 2021, doi:10.3390/jcm10215092_

Round 1

Reviewer 1 Report

This article reports results of a multicenter japanese observational study, assessing the effectiveness of non invasive positive pressure ventilation on short-term clinical outcomes of patients hospitalized for acute decompensated heart failure.

Broad comments

This article is well written and easy to read. It gives interesting results, but needs a better contextualisation and more statistical rigor.

My first comment concerns the lack of clarity and preciseness of your question. You say very shortly that you are interested in older subjects, but 1/ you do not exploit the literature very far in this direction 2/ you don’t justify specific subgroup analyses to assess this question 3/ you do not compare, in the discussion, the effect sizes in your study with effect sizes in meta-analyses, to explain it by differences in case mix (except for mortality rate). This is a weakness of your paper, because it is indeed the interest of observational studies, comparatively to randomized trials. So you could demonstrate it better.

My second comment concerns the use of the propensity score as a matching score. This is generally a good method, but in your case you exclude 80% of patients of the pre-matched cohort, which induces a selection bias. You should perform a sensitivity analysis using IPTW or adjustment on propensity score.

My third comment concerns your subgroup analyses in the post-matched cohort. I think that they are very dangerous : the selection bias due to the matching may induce confounding bias. Here also I would recommend to perform these analyses using the pre-matched cohort. Furthermore, your subgroup analyses do not follow good practices. You did not adjust type I error for multiple testing (same remark stands for your multiple outcomes) ; you perform stratified analyses even if the interaction test is not significant ; you do not present descriptive results ; and see first comment : your subgroups are not justified.

Specific comments

Line 39 « Management in the acute decompensated phase has been the cornerstone of HF treatment. [2] » This statement is overstated, the reference does not support it and is of very low quality.

Line 48« The small randomized control trials (RCTs) exploring the impact of NPPV on the outcomes have been conducted by different research groups. [8–11] ». References 8-10 are 3 of the most recent meta-analyses on the hypothesis. Why don’t you present them as such, rather than as small RCTs ? Why do you cite a 30-years old study (11) ?

Line 52 « but those have been inconsistent regarding its impact on in-hospital mortality. [8,9] ». I do not agree :

  • Berbenetz 2019 Cochrane : RR = 0.65
  • Weng 2010 Ann intern med : RR = 0.64
  • Masip 2005 JAMA : RR = 0.55.

The impact is consistant ; problems are rather uncontrolled biases due to the lack of description of standard procedures (diuretics, intubation criteria …) in this context of open-label studies.

L.59 Litterature gives some inputs about which patients could most benefit from NPPV (not only age, but also coronary disease, hyper/eucapnia …). As this is an important question in your article, and moreover because you excluded patients with acute coronary syndrome, you should developp this further.

L71 It is not clear wether patients with acute coronary syndrome were excluded from WET-HF or from your specific analysis.

L133 Did you exclude variables representing events occurring after the NPPV decision ? In your discussion, you mention that you did not know in which setting occurred the NPPV ; how can you be sure that in-hospital medications preceeded NPPV ? If you’re not sure, there is a risk of protopathic bias, and you should not use these variables.

Please use less abbreviations in tables, which are difficult to read.

Table 1 : typo error in % of HFrEF. BMI lacks in this table.

Table 2 : reference value of etiology is unclear. Did you assess log-linearity of the two classes variables that you analyze as continuous variables (admission year ; BNP or Nt-proBNP quartile) ?

L.153 I do not think that « third trimester » can mean « third admission period » in this case. It means July-August-September.

Descriptive results leading to figure 3 should be shown in an appendix (number of subjects and event rates in each subgroup, OR and IC).

L.281, reference to table 1 is false. You mean figure 2 ?

Author Response

- Reviewer 1

We thank the Reviewer for his/her thoughtful comments.  Below are our responses:

Broad comments

This article is well written and easy to read. It gives interesting results, but needs a better contextualisation and more statistical rigor.

  1. My first comment concerns the lack of clarity and preciseness of your question. You say very shortly that you are interested in older subjects, but 1/ you do not exploit the literature very far in this direction 2/ you don’t justify specific subgroup analyses to assess this question 3/ you do not compare, in the discussion, the effect sizes in your study with effect sizes in meta-analyses, to explain it by differences in case mix (except for mortality rate). This is a weakness of your paper, because it is indeed the interest of observational studies, comparatively to randomized trials. So you could demonstrate it better.

Response: We appreciate the Reviewer for his/her valuable comments. We sought to correctly cite the literature and amended some references . Please see the Response to the comments #4, #5 and  #7. Regarding the subgroup analysis, please see the Response to the Comment #3. Regarding the effect size, the comments were added in the Discussion section, which is shown lines 309 - 312.

  1. My second comment concerns the use of the propensity score as a matching score. This is generally a good method, but in your case you exclude 80% of patients of the pre-matched cohort, which induces a selection bias. You should perform a sensitivity analysis using IPTW or adjustment on propensity score.

Response: According to the Reviewer’s comments, we additionally conducted logistic regression analysis for each endpoint with adjustment for propensity score using the pre-matched cohort.  Multiple regression analysis was conducted to see the association of NPPV use with LOS with adjustment for calculated propensity score in the pre-matched cohort. NPPV use was associated with lower ETI rate and longer LOS. This description was added in lines 173-178 in the Methods section, lines 258-263 and new Table 3 in the Results section .

  1. My third comment concerns your subgroup analyses in the post-matched cohort. I think that they are very dangerous : the selection bias due to the matching may induce confounding bias. Here also I would recommend to perform these analyses using the pre-matched cohort. Furthermore, your subgroup analyses do not follow good practices. You did not adjust type I error for multiple testing (same remark stands for your multiple outcomes) ; you perform stratified analyses even if the interaction test is not significant ; you do not present descriptive results ; and see first comment : your subgroups are not justified.

Response: According to the Reviewer’s comments we conducted propensity matching for each stratified group in the pre-matched cohort and re-analyzed. And due to the issue of multiple testing, we omitted the findings of cardiac death. The findings are shown in Figure 3.

By a stratified analysis we sought to determine the patient population that would benefit from the use of NPPV. Older age, hypotension, malnutrition, and history of ADHF hospitalization are risk factors for adverse outcome in patients with HF. Thus, in addition to basic information such as gender, BMI, and LVEF, the patients were stratified by the factors potentially affecting the effect of NPPV use such as age, SBP, nutritional status and history of ADHF hospitalization. This description was added in lines 163-172.

Specific comments

  1. Line 39 « Management in the acute decompensated phase has been the cornerstone of HF treatment. [2] » This statement is overstated, the reference does not support it and is of very low quality.

Response: According to the Reviewer’s comments, we corrected the text and the reference was changed to another one, which is shown in lines 41-46.

  1. Line 48« The small randomized control trials (RCTs) exploring the impact of NPPV on the outcomes have been conducted by different research groups. [8–11] ». References 8-10 are 3 of the most recent meta-analyses on the hypothesis. Why don’t you present them as such, rather than as small RCTs ? Why do you cite a 30-years old study (11) ?

Response: We appreciate the Reviewer for his/her helpful advice. We changed the references to the previously conducted randomized control studies which enrolled relatively larger number of patients (ref #6-9). We removed very old study (ref #11 in the original manuscript) from the reference.

  1. Line 52 « but those have been inconsistent regarding its impact on in-hospital mortality. [8,9] ». I do not agree :

Berbenetz 2019 Cochrane : RR = 0.65

Weng 2010 Ann intern med : RR = 0.64

Masip 2005 JAMA : RR = 0.55.

The impact is consistant ; problems are rather uncontrolled biases due to the lack of description of standard procedures (diuretics, intubation criteria …) in this context of open-label studies.

Response: We appreciate the Reviewer for his/her valuable comments. We completely agree to the comments made by the Reviewer. Actually, the findings of the RCTs showed the mixed results but the meta-analysis showed the consistent findings. We revised the description in lines 72-81 in the Introduction section and lines 315-324 in the Discussion section.

  1. L.59 Litterature gives some inputs about which patients could most benefit from NPPV (not only age, but also coronary disease, hyper/eucapnia …). As this is an important question in your article, and moreover because you excluded patients with acute coronary syndrome, you should developp this further.

Response: One small RCT has shown the beneficial effect of NPPV in patients with acute myocar-dial infarction-associated pulmonary edema that is not complicated by shock. The description of this article was added in addition to those of the limitations in the previously conducted studies, which is shown in lines 78-81.

  1. L71 It is not clear wether patients with acute coronary syndrome were excluded from WET-HF or from your specific analysis.

Response: The patients with ACS were not registered in the WET-HF registry. The description was amended, which is shown in lines 88-89.

  1. L133 Did you exclude variables representing events occurring after the NPPV decision ? In your discussion, you mention that you did not know in which setting occurred the NPPV ; how can you be sure that in-hospital medications preceeded NPPV ? If you’re not sure, there is a risk of protopathic bias, and you should not use these variables.

Response: We appreciate the Reviewer for his/her important suggestions. We deleted in-hospital treatment such as carperitide, nitrates and catecholamine use from this model and reconstructed it. Those data are shown in Table 2. Due to this amendment, the deviation of in-hospital treatment between the 2 groups remained even in the post-matched cohort. This description was added in the Limitation, which is shown in lines 366-369.

  1. Please use less abbreviations in tables, which are difficult to read.

Response: We spelled out some of abbreviations in the Tables.

  1. Table 1 : typo error in % of HFrEF. BMI lacks in this table.

Response: The errors were corrected.

  1. Table 2 : reference value of etiology is unclear. Did you assess log-linearity of the two classes variables that you analyze as continuous variables (admission year ; BNP or Nt-proBNP quartile) ?

Response: We corrected these variables to ordinal variables and reconstructed the model of liner regression analysis for propensity score calculation, which is shown in Table 2.

  1. L.153 I do not think that « third trimester » can mean « third admission period » in this case. It means July-August-September.

Response: We corrected the errors, which are shown in line 218 and Supplementary Results.

  1. Descriptive results leading to figure 3 should be shown in an appendix (number of subjects and event rates in each subgroup, OR and IC).

Response: According to the Reviewer’s comments, the number of subjects and event rates on each subgroup, odds ratio and 95% confidence interval were added in Figure 3.

  1. L.281, reference to table 1 is false. You mean figure 2 ?

Response: The error was corrected.

Reviewer 2 Report

In this paper, the authors sought to assess the impact of NPPV on the prognosis of patients hospitalized for decompensated HF.

The  paper includes a large population of patients, nevertheless, I have several issues that need to  be solved and that are listed below:

1) I found several mistakes in the English form of the paper. The paper should be reviewed by a native English speaker.

2) In Table 1 the authors should display only data of the 2 matched cohorts. The comparison of the 2 cohorts before matching is not necessary to the design and purpose of the study. Similarly, the paragraph describing the comparison between the 2 non-matched cohorts is not useful and should be erased.

3)  In Table 1 the authors should indicate data on BNP or NT-proBNP. Displaying data on both these parameters is redundant.

4) In Table 1, a specific heading should precede the description of echo variables.

5) In Table 2 the authors indicate Admission year (2014-2017 / 2010-2013 / 2006-2009) as a variable. Nevertheless, they should indicate which time is the reference for statistical analysis.

6) Before providing the results of a multivariable analysis (Table 2), the authors should include data on univariable analysis and describe how data were inserted in the multivariable model.

7) The authors should specify if the data depicted in Figure 3 correspond to the result of a univariable or multivariable analysis.

8) It is clear from the paper that NPPV is associated with a longer in-hospital stay. Nevertheless, I think that the data displayed in table 2 are not relevant for the full comprehension of the paper and can be erased.

9) In the discussion the authors state that: “In the present study, ischemic aetiology and higher SBP were associated with NPPV 284 use (Table 2)”. Actually, Table 2 displays a comparison between groups. This comparison shows that patients undergoing NPPV had a higher prevalence of ischemic cardiomyopathy and higher systolic blood pressure level. These results do not mean that there is a statistical association between SBP and NPPV use.

Author Response

-Reviewer 2

We thank the Reviewer for his/her thoughtful comments.  Below are our responses:

  • I found several mistakes in the English form of the paper. The paper should be reviewed by a native English speaker.

Response: The original manuscript was edited by the English editor in the English editing service . The certificate was attached. And once again the revised manuscript was thoroughly checked at this time and typos were corrected.  

  • In Table 1 the authors should display only data of the 2 matched cohorts. The comparison of the 2 cohorts before matching is not necessary to the design and purpose of the study. Similarly, the paragraph describing the comparison between the 2 non-matched cohorts is not useful and should be erased.

Response: According to the Reviewer’s comments,  the baseline characteristics of the pre-matched cohort was removed and moved to the Supplement, the description of the pre-matched cohort in the text was also removed and moved to the Supplement. The summary of the findings in the pre-matched cohort in the Discussion was also removed.

  • In Table 1 the authors should indicate data on BNP or NT-proBNP. Displaying data on both these parameters is redundant.

Response: Since either one of BNP or NT-proBNP was measured at admission for each patient, we were concerned if the presentation of just one of these might lead to bias. We are very grateful if the Reviewer could understand the circumstance that we need to present both of them.

4) In Table 1, a specific heading should precede the description of echo variables.

Response: A heading “Echocardiography” was added in Table 1.

5) In Table 2 the authors indicate Admission year (2014-2017 / 2010-2013 / 2006-2009) as a variable. Nevertheless, they should indicate which time is the reference for statistical analysis.

Response: We corrected handling of Admission Year from continuous variable to ordinary one and clearly indicated which one is reference. 

6) Before providing the results of a multivariable analysis (Table 2), the authors should include data on univariable analysis and describe how data were inserted in the multivariable model.

Response: The results of univariable analysis were added in Table 2.  The variables were selected a priori for their potential to be strongly associated with NPPV use. For the multivariable analysis, age, sex, LVEF and the variables from those that showed an association with NPPV use in the univariable analysis with p value <0.1 were employed. This description was added in lines 152-155.

7) The authors should specify if the data depicted in Figure 3 correspond to the result of a univariable or multivariable analysis.

Response: Figure 3 illustrates the stratified analysis depicting the association of NPPV use and respective endpoint and these were all the findings of univariable analysis. The finding of “overall” population corresponds to the findings presented in Figure 2 (univariable analysis). Event rates were also specified in Figure 3. For this revised manuscript, the pre-matched cohort was stratified and then propensity matching was conducted for each stratified group according to the Reviewer 1’s comments.

8) It is clear from the paper that NPPV is associated with a longer in-hospital stay. Nevertheless, I think that the data displayed in table 2 are not relevant for the full comprehension of the paper and can be erased.

Response: In this revision we totally changed the strategy for stratified analysis according to the Reviewer 1’s comments. The pre-matched population was initially stratified and then propensity matching was conducted. Further, we additionally stratified the population  by nutritional status (CONUT score). The subgroups related to frailty were associated with significantly longer LOS. Although the differences between the subgroups were small, but that in females, non-ischemic etiology, and CONUT score >3 are statistically significant.  We believe these data provide some implication on which patients can benefit more or less from NPPV. However, according to the Reviewer’s comments, we removed this Table and moved to Supplement.

9) In the discussion the authors state that: “In the present study, ischemic aetiology and higher SBP were associated with NPPV 284 use (Table 2)”. Actually, Table 2 displays a comparison between groups. This comparison shows that patients undergoing NPPV had a higher prevalence of ischemic cardiomyopathy and higher systolic blood pressure level. These results do not mean that there is a statistical association between SBP and NPPV use.

Response: Whereas Table 1 shows the comparison between NPPV and non-NPPV groups, table 2 indicated the logistic regression analysis depicting the association of variables with NPPV use. According to the Comment #2, these descriptions were removed from the text.

Round 2

Reviewer 1 Report

Thank you for these answers, the paper improved. There are still some pending comments.

Stratified analysis

I do not know if you misunderstood my advice or if you disagree … I think that propensity scores are not a good idea to assess effect of interventions in subgroups. You should perform a classical multivariable analysis, with interaction terms, and then perform stratified analyses only in strata corresponding to the significant interaction term. Where the interaction term is non significant, you should only present descriptive analyses.

With your current method, you do not perform interaction tests, which is a flaw. Therefore, you cannot state that « stratified analysis clearly demonstrated the subgroups that showed advantages and/or disadvantages along with receiving NPPV in a real-world setting ». Replace by « stratified analysis clearly suggest the subgroups that showed advantages 308 and/or disadvantages along with receiving NPPV in a real-world setting » (L.308)

Reference values of categorical variables of table 2

Please present one row for reference value, with OR = 1, 95% CI = -, p=-, and one row for each other value. For etiology, how can you have two reference values ?

Effect size

Effect size is not sample size. Effect size is the RR, or the risk difference, or any measure that tells you how much health you gain for your patients. We do not care of a statistically significant difference if it concerns a difference of 1 day in length of life. When interpreting results in clinical research, you need to comment the effect size, id est the clinical significance, along with the statistical significance.

L218

Phrase without verb, is it what you meant ?

Author Response

Stratified analysis

I do not know if you misunderstood my advice or if you disagree … I think that propensity scores are not a good idea to assess effect of interventions in subgroups. You should perform a classical multivariable analysis, with interaction terms, and then perform stratified analyses only in strata corresponding to the significant interaction term. Where the interaction term is non significant, you should only present descriptive analyses.

With your current method, you do not perform interaction tests, which is a flaw. Therefore, you cannot state that « stratified analysis clearly demonstrated the subgroups that showed advantages and/or disadvantages along with receiving NPPV in a real-world setting ». Replace by « stratified analysis clearly suggest the subgroups that showed advantages 308 and/or disadvantages along with receiving NPPV in a real-world setting » (L.308)

Response: We apologize for our misunderstanding. According to the Reviewer’s comments, we have re-analyzed the data and reformatted the manuscript accordingly. These data are shown in new Figures 3C (ETI rate) and 3D (in-hospital mortality), and new Table 3B (LOS), and they have followed your suggestions.

Further, we have edited the description in line 302 according to the Reviewer’s comment. Once again, we thank the Reviewer for his/her thoughtful comments.

Reference values of categorical variables of table 2

Please present one row for reference value, with OR = 1, 95% CI = -, p=-, and one row for each other value. For etiology, how can you have two reference values ?

Response: We amended the expression according to the Reviewer’s comment. For etiology, we set ICM as reference.

Effect size

Effect size is not sample size. Effect size is the RR, or the risk difference, or any measure that tells you how much health you gain for your patients. We do not care of a statistically significant difference if it concerns a difference of 1 day in length of life. When interpreting results in clinical research, you need to comment the effect size, id est the clinical significance, along with the statistical significance.

Response: We apologize for the misunderstanding on our part. Following the Reviewer’s comments, we added the below statement to the discussion section.

In the present study a risk ratio for ETI in the propensity match analysis was 0.515 and an estimated risk ratio calculated from OR in the multivariable regression analysis by the formula by Viera AJ (South Med J 2008;101(7):730-4) was 0.480. From these findings the effect sizes in our study were consistent with the recent Cochrane meta-analysis (risk ratio 0.49).

This description is shown in lines 316-319.

L218

Phrase without verb, is it what you meant ?

Response: The error was corrected, which is shown in lines 188-189.

Reviewer 2 Report

The current version of the paper is significantly improved compared to the previous one. Nevertheless, I have some remaining issues:

  • Lines 218-221. The phrase: “Admission in the third trimester admission period (2014-2017), is-218 chemic aetiology, higher systolic blood pressure (SBP), and parameters representing severe 219 ADHF such as lower oxygen saturation (SpO2), NYHA [..]” is without a verbal form.
  • In the headings of Table 1, the authors indicate that they are depicting the pre-matched and post-matched cohort. I guess that data in Table 1 correspond to the post-matched cohort.
  • In Table 3 the authors show the multivariable regression analysis for in-hospital outcomes with adjustment for propensity score in the 

Author Response

We thank the Reviewer for his/her thoughtful comments.  Below are our responses:

Lines 218-221. The phrase: “Admission in the third trimester admission period (2014-2017), is-218 chemic aetiology, higher systolic blood pressure (SBP), and parameters representing severe 219 ADHF such as lower oxygen saturation (SpO2), NYHA [..]” is without a verbal form.

Response: The error was corrected, which is shown in lines  188-189.

In the headings of Table 1, the authors indicate that they are depicting the pre-matched and post-matched cohort. I guess that data in Table 1 correspond to the post-matched cohort.

Response: The error was corrected.